# Study of Polonium (^210^Po) Activity Concentration in Fruit Wines Derived from Different Locations in Poland

**DOI:** 10.3390/molecules28010438

**Published:** 2023-01-03

**Authors:** Paweł Rudnicki-Velasquez, Alicja Boryło, Marcin Kaczor, Jarosław Wieczorek, Jarosława Rutkowska

**Affiliations:** 1Institute of Human Nutrition Sciences, Faculty of Human Nutrition, Warsaw University of Life Sciences (WULS-SGGW), Nowoursynowska St. 159c, 02-776 Warsaw, Poland; 2Faculty of Chemistry, University of Gdańsk, Wita Stwosza Str. 63, 80-308 Gdańsk, Poland

**Keywords:** polonium-210, fruit wines, annual effective dose, radioactivity, geographic provenance

## Abstract

This study aimed at assessing the activity concentration and the annual effective dose of polonium-210 (^210^Po) in fruit wines derived from four locations in Poland (Warmian–Masurian, Podlaskie, Lubelskie and Małopolskie voivodeships). The fruit wines differed significantly (*p* < 0.05) in ^210^Po activity depending on the production site, with the Małopolskie site having the highest activity (61.4–221.4 mBq/L) and the Podlaskie having the lowest (3.5–97.1 mBq/L). The site differentiation was due to environmental conditions—soil parameters (uranium concentration), precipitations and terrain characteristics, e.g., the proximity of the lakes. The increased activity concentration of ^210^Po in samples from Małopolska compared with the other sites probably derived from the environment polluted with aqueous wastes and particulate air pollution. The annual effective dose due to the ingestion of fruit wines ranged from 0.112 to 1.214 µSv/year. These levels of exposure are safe according to the WHO criterion (0.1 mSv per year for ingestion) and to the IAEA reference level for public exposure including food (1 mSv per year). Summing up, the data obtained provide information on the activity concentration of ^210^Po in fruit wines and increase databases on the natural radioactivity of foodstuffs. Future work is needed to examine ^210^Po activity in samples from all vineyard regions in Poland.

## 1. Introduction

Natural radiation comes from cosmic rays, naturally occurring radioactive elements in the Earth’s crust, and from the radioactive decay of primordial radionuclides such as the ^40^K, ^232^Th and ^238^U series [1,2,3]. Radionuclides undergo natural fallout with rain and oceans. Additionally, nuclear facilities may emit certain amounts of them [4].

Polonium-210 (^210^Po) is a radioactive isotope and is a product of the ^238^U decay chain. ^210^Po with a half life of 138 days is an α-emitter, and the radiation energy is equal to 5.3 MeV. The energy released by the decay is so large that 0.5 g of polonium will self-heat to a temperature above 500 °C and have a blue glow [5,6,7,8]. ^210^Po thus has a relatively high radioactivity which corresponds to its very small quantity and is considered one of the most radiotoxic natural radioactive isotopes known. About 0.05 μg of the radionuclide is considered a deadly dose [9]. A fatal oral dose of ^210^Po is much lower than the acute lethal dose of hydrogen cyanide in humans (LD_50_ = 200–300 mg) [5,10].

Polonium-210 is radiotoxic only if taken internally [11]. According to the model recommended by the International Commission on Radiological Protection (ICRP), about 10–50% of ingested ^210^Po is absorbed by the intestine and deposited mostly in the liver, kidneys, spleen, red bone marrow and in other tissues [6,12,13]. A study by Henricsson et al. [9] reported that there was no apparent gender dependence in gastrointestinal uptake, and no apparent dependence on the chemical form of the polonium ingested.

The absorbed ^210^Po is concentrated initially in red blood cells and results in reducing the number of lymphocytes, which in turn decreases the immune response with a highly increased sensitivity for infections [5]. Ingested ^210^Po is more readily absorbed than some other α-emitting radionuclides, e.g., plutonium-239 [10]. It should also be noted that the alpha particles emitted from the decay of ^210^Po revealed a greater relative biological effectiveness considering cell killing or cancer induction than gamma- or X-rays [6].

^210^Po can be easily transported across the body while remaining undetectable as the high-energy alpha particles are characterized by a short range and can be blocked by quite a thin barrier, e.g., skin, meaning the Geiger counter is unable to detect the radiation emitted by ^210^Po [14]. Early symptoms of ^210^Po poisoning are indistinguishable from those of a wide range of chemical toxins. Hence, the diagnosis can be delayed and even missed without a high degree of suspicion [14]. Consequently, due to its specific activity (1.66 × 10^14^ Bq/g) and properties, ^210^Po is considered the most dangerous radionuclide for human beings [8,14,15].

Human beings are steadily exposed to ^210^Po, which occurs at low concentrations in the environment as part of the uranium decay [14]. Previous studies revealed that ^210^Po was an important contributor to human ionizing radiation exposure through food [11]. A study on establishing radionuclide activity concentrations in a New Zealand diet revealed the determination of 210-Po activity in 103 of 160 analyzed samples that encompassed a large variety of the foods [16]. The research of Meli et al. [17] revealed that an effective dose from ^210^Po ingested by the total diet accounts for 5–12% of the natural radiation exposure in Italy.

Studies on the activity concentration of natural radionuclides in food revealed that seafood (fish, shellfish, crustaceans) contained higher activity than other food categories [16,17,18]. Seafood made the most significant contribution to the dietary intake of ^210^Po because of its high affinity for proteins [11]. However, a substantial ^210^Po activity concentration was also assayed in leafy vegetables [17]. A more recent study by Wang et al. [19] revealed that one of the main reasons for the contamination of leafy vegetables by the ^210^Po radionuclide was crops cultivation close to the coal mines and other mining activities.

Trdin and Benedik [20] reported that the consumption of infant milk formulas was responsible for a ^210^Po radiation dose ranging from 230 to 350 µSv y^−1^. Although the assessed dose from ^210^Po ingestion with milk formulas was under 1 mSv y^−1^, which is the level that is allowed for public exposure, it should be pointed out that these products are offered for the segment of the population that is the most sensitive to radiation hazards [20,21].

Previous studies revealed the contamination of drinking water with naturally occurring β- and α-radionuclides including ^210^Po [22,23,24,25]. Generally, the annual effective dose of ^210^Po ingestion via drinking water in Italy, Poland and Hungary indicated an acceptable carcinogenic risk for the populations [22,23,24,25]. Additionally, Kurttio et al. [26] concluded that even though radionuclides ingested with water from drilled wells were a source of radiation exposure, they were not associated with a substantially increased risk of bladder or kidney cancers in concentrations occurring in drilled wells [26]. Significant regional differences in ^210^Po activity concentrations in drinking water in Italy (0.13–11.3 mBqL^−1^) and in Poland (0.35–3.43 mBqL^−1^) are to be noted [24,25]. Higher than in other countries, the activity concentrations of ^210^Po in the drinking water from the Langat River area in Malaysia were due to the weathering process on granitic formations of the river basin and also might be the result of mining, agriculture and industrial activities [27].

Another study revealed the contamination of wine with various radionuclides: ^137^Cs; ^234^U; ^238^U; ^87^Sr/^86^Sr, the strontium isotope ratio; and ^210^Po [2,28,29,30,31]. Variations in the ^87^Sr/^86^Sr isotope ratios in wine were used to differentiate the geographic origin in wine-producing countries [29]. It should be pointed that the Sr ion is toxicologically undesirable because of its ability to substitute calcium in bones, thereby possibly impacting bone density [32]. The activity concentration of ^210^Po in Italian red wine samples from 16 regions ranged greatly (19.5–223 mBq/L^−1^), probably due to differences in the fallout of the atmospheric polonium on the leaves the vines and on the grapes between regions [2].

There are many sources of radionuclides in grapes and hence in wine. Radionuclides are transported to wine due to fruit contamination by their deposition on the fruit surface, absorption through the skin, transport to the pulp, deposition to the soil, root uptake and transfer to the fruit [33]. The concentrations of most radionuclides in grapevine fruit depend on the presence of a given nuclide in the soil of the vineyard. Soil fertilization and weather conditions throughout the annual growing season affected concentrations of the radioactive elements in wines [34]. The recent study of Gulan et al. [35] revealed that the absorption, bioaccumulation and distribution of radionuclides in plants are affected by many factors (such as soil properties, vine types, local climate, etc.) and cannot be readily predicted from the radioactivity content in the underlying soil. The leading source of ^210^Po in plants, however, is still dry and wet atmospheric fallout [6].

According to EU legislation (Reg. (EU) 1308/2013), wine is defined as a “product obtained exclusively from the total or partial alcoholic fermentation of fresh grapes, whether or not crushed, or of grape must” [36]. On the other hand, the interest in fruit wines produced from fruits other than grapes has recently been increasing in many countries [37]. Poland belongs to the countries in which wine consumption is becoming increasingly popular; hence, the number of vineyards has substantially increased (from 20 in 2010 to as much as 380 in 2020) [38]. The production of wine in Poland is still developing and is concentrated mainly in small local manufactures, and new wine regions are emerging. Wine production in Poland is based not only on locally grown grapes but it also utilizes the potential of many varieties of fruits (e.g., black currant, strawberry, chokeberry, apple, cherry). However, to date, the information on radionuclides in fruit wines is scarce, and to the best of our knowledge, this is the first report on ^210^Po activity concentrations in locally manufactured fruit wines.

Thus, the aim of this study was to determine the activity concentration of ^210^Po in locally manufactured fruit wines derived from diverse locations in Poland, together with the radiation dose due to ^210^Po ingestion with the wine.

## 2. Results and Discussion

### 2.1. Assessment of ^210^Po Activity Concentrations in Fruit Wine Samples

Polonium-210 is a significant radionuclide in the ^238^U decay series due to its high radiotoxicity [39]. Thus, it is worth noticing that ^210^Po activity concentration was detectable in all 86 analyzed fruit wine samples derived from four locations in Poland, ranging from 3.49 to 221.44 mBq/L. That range was arbitrarily split into four categories as follows: 1: <10 mBq/L; 2: 10–100 mBq/L; 3: 101–200 mBq/L; and 4: >200 mBq/L (Figure 1).

Considering the regional mean values, only those from the Małopolskie and Lubelskie voivodeships did not differ significantly (*p* ≤ 0.05). However, the ^210^Po activity concentration in the samples from the Małopolskie voivodeship exceeded 60 mBq/L and ranged from 61.4–221.4 mBq/L (Table 1). 

This may have resulted from a higher level of uranium in the soil in Małopolska than in the other three sites: 2.5–3.4 mg U/kg soil and 0.5–1.6 mg U/kg soil, respectively (Table 2). As reported by Giri et al. [3], natural radionuclides entering the food chain are mostly derived from the soil and, in effect, variation in the soil radionuclide content is a prime source of geographic variability.

It should also be noticed that the five samples from the Małopolska region contained more than 130 mBq/L, and the other two ones contained more than 200 mBq/L (Figure 1). A predominant area in the Małopolska voivodeship was contaminated by Silesian coal mines, which produce large volumes of waste waters containing uranium series isotopes directly discharged into the Vistula river [41]. Another study also highlighted the problem of exceeding the acceptable levels of carcinogenic Cd in wild fruits from the Małopolska voivodeship [42]. The study by Skwarzec et al. [25] revealed that increased activity concentrations of ^210^Po in drinking water from Głogów in Poland was due to the contamination of the Oder river with wastes of copper smelter. The Małopolskie voivodeship was also mostly contaminated by higher particulate air pollution, as reflected in the higher amounts of coarse particles (PM_2.5_) and fine particles (PM_10_; Table 2) compared with the other locations. Additionally, heavy metals and other toxic compounds, PM_2.5_ and PM_10_, contain radionuclides. The radioactivity of the coarse particles derived mainly from the radon progeny are in disequilibrium between ^210^Pb and ^210^Po [43].

Despite the lower level of uranium in the soil of Lubelskie than in the Małopolska site, wine samples from the Lubelskie voivodeship contained a relatively high activity concentration of ^210^Po (Table 1). Seven of the twenty-three samples from Lubelskie contained more than 80 mBq/L of ^210^Po activity concentrations (Figure 1). The substantial abundance of Polonium-210 in those samples may have derived from locally increased anthropogenic activities, such as phosphate or processing (coal-fired power stations, metal smelting) [44]. Additionally, the ^210^Po activity in the wines may have derived from a fertilizer plant located in the Lubelskie site as they process a phosphate rock whose waste is phosphogypsum, which is contaminated with uranium isotopes including the ^210^Po radionuclide [45].

Generally, most of the samples from the Warmian–Masurian voivodeship contained ^210^Po activity in the range 10–100 mBq/L. However, in three samples, the ^210^Po activity concentration exceeded 100 mBq/L (Figure 1, Table 1). The respective wine manufactures are situated between two big lakes. Probably, the proximity of the lakes in this particular site contributed to the higher ^210^Po activity concentration in the wines, as compared with the other samples from the Warmian–Masurian voivodeship. The study by Nelson et al. [39] indicated that the substantial levels of ^210^Po in the lake are a natural phenomenon and are likely unrelated to waste water treatment discharges.

Our study also revealed significant (*p* ≤ 0.05) differences in the activity concentrations of ^210^Po between wines from two neighboring sites (Warmian–Masurian and Podlaskie). Ten of the twenty analyzed samples from the Podlaskie voivodeship contained less than 10 mBq/L of ^210^Po activity (Figure 1). So, low levels of ^210^Po were not found in any of the samples from Warmian–Masurian (or in samples from the remaining two location). As was stated in the study by Desideri et al. [2], polonium contamination is a direct one, as it is due to radionuclide particle deposition from the atmosphere onto the grape. Thus, in our opinion, differences in the activity concentrations of ^210^Po between wines from two neighboring production sites could stem from the microclimate of these two voivodeships. Higher activity concentrations of ^210^Po in the fruit wine from the Warmian–Masurian voivodeship than in the samples from the Podlaskie voivodeship may have been due to differences in weather conditions. The Warmian–Masurian voivodeship is characterized by a higher annual precipitation than Podlaskie (about 100 mm higher; Table 3). It should be also mentioned that nearly 30% of the Warmian–Masurian voivodeship area is covered by lakes, which may be a source of ^210^Po [39]. Both those factors may facilitate the accumulation of isotopes in fruits and wines [27].

Like in our study, the ^210^Po activity concentration in fruit wines were assayed in Italian wines produced from grapes [2]. The study by Desideri et al. [2] also revealed that the activity concentrations of ^210^Po in wines were region dependent. In that study, a lower activity concentration of ^210^Po was found in wine samples from the Liguria region (12 mBq/L) and a higher one in samples from the Veneto region (223 mBq/L) [2].

The activity concentration range of ^210^Po in fruit wines from the Podlaskie voivodeship (3.5–97.1 mBq) was similar to that assessed in fruit vegetables (tomato, cucumber, zucchini) and in fruits from Italy and New Zealand [16,17,46] but was higher in the other three sites. Except for samples from the Podlaskie voivodeship, wines had higher ^210^Po activity concentrations than basic components of the human diet (milk, meat and their products) [16,17,46].

Some studies exposed a high contamination of food products with ^210^Po, e.g., seafood, especially shellfish, leafy vegetables and infant formulas [17,20,46,47,48]. Fruit wines from two locations (Małopolska and Lubelskie) were similarly contaminated with ^210^Po radionuclides as leafy vegetables, as analyzed in Italy, Vietnam and reported by the United Nations Scientific Committee on the Effects of Atomic Radiation (UNSCEAR) [17,46,47]. Additionally, wine samples from these two locations contained similarly high ^210^Po activity concentrations, as found in infant formulas in Slovenia [20]. Irrespective of provenance, ^210^Po is now regarded as the most dangerous radiation hazard involved with food consumption, yet the ^210^Po contamination of fruit wine samples was much lower than that of shellfish from Europe and New Zealand [17,46,48].

The UNSCEAR report and literature data do not provide information about radionuclides in fruit wines; thus, the results obtained in our study represent an important complement of the European data of natural radioactivity foodstuffs [46].

### 2.2. Radiation Dose Estimation

The calculated annual effective doses due to the ingestion of ^210^Po in fruit wines derived from different locations for adults is presented in Table 3. The highest doses of ^210^Po came from the consumption of wine from the Małopolskie and Lubelskie voivodeships: from 0.70 to 2.572 μSv/year and from 0.443 to 1.930 μSv/year, respectively. The calculated annual effective dose due to the consumption of wine from Warmian–Masurian and Podlaskie was much lower and ranged from 0.112 to 1.430 μSv/year and from 0.041 to 1.108 μSv/year, respectively (Table 3).

The exposure of adults to ionizing radiation from the ingestion of ^210^Po in Italian wines was higher than in our study (0.258–48.8 μSv/year) [2] due to a higher wine consumption in Italy than in Poland—0.5 L per day vs. 9.5 L/year, respectively [2]. Meli et al. [17] reported that the consumption of basic food products (milk, meat, grain, roots vegetables) contributed to the radiation dose due to ^210^Po ingestion ranging 2.1–4.5 μSv/year, which was about 2–3 fold higher than the effective dose from fruit wine intake estimated in our study [Table 3]. More recent studies highlighted that the consumption of seafood, especially shellfish in New Zealand, and the consumption of water spinach in Vietnam were the highest contributors to human exposure to the ^210^Po isotope [11,47].

The estimated levels of the annual effective dose due to the ingestion of fruit wines are arguably relatively safe compared with WHO guidelines for specific radionuclides having a generic criterion of 0.1 mSv per year for ingestion and with the ICRP reference level for public exposure in specific existing exposure situations including food—1 mSv per year, which was not exceeded in any wine sample [49].

The contribution of fruit wines to the annual effective dose in Poland is small compared with that resulting from the ingestion of ^210^Po and ^210^Pb together with basic food products and water (54 μSv for both radionuclides) [50]. It should be noticed that the dose contribution via ^210^Po ingestion represents only a partial dose of the radionuclides considered and does not represent the real total radiation due to wine ingestion. It should be noted that the long-term effects of a low dose of polonium exposure are unknown. As indicated by Seiler [51], much of the important biological and toxicological research on ^210^Po is more than four decades old. Thus, there is a need to conduct new research using modern laboratory tools which will included biological and epidemiological investigations conducted in ^210^Po-contaminated areas [51].

## 3. Materials and Methods

### 3.1. Sample Collection

Fruit wine samples (86 samples in total) were collected from small wine manufacturers located in four voivodeships in Poland: Warmian–Masurian, Podlaskie, Lubelskie and Małopolskie (Figure 1). The wines were produced from one type or from various types of locally grown fruits: raspberries, apples, black currants, cherries, plums, chokeberries and red currants.

Detailed characteristics of fruit wine samples are presented in Appendix A. The environmental conditions and specific characteristic of the production voivodeship are presented in Table 1. All the wine samples were collected between October and December 2021.

### 3.2. Alpha Spectrometry Analysis

The volume of the wine was measured, and the recovery index in the form of ^209^Po with T1/2 = 124 y and decay particles energy of 4.979 MeV was added [52]. The results were obtained from 2 samples analyzed in parallel. The obtained samples were selected according to the rules presented in the Introduction. The volume of the wine in the sample ranged from 0.5 to 1 L. After measuring the volume, the sample was evaporated to dryness. Concentrated (63%) HNO_3_ (100 mL) was added to the wine, and then it was slowly evaporated under constant mixing. When adding acid, it was necessary to add it very slowly, especially to the wines with a high sugar content, due to a violent reaction. After evaporation, another 200–300 mL of concentrated HNO_3_ was added to finally mineralize the residue. After mineralization, the samples were evaporated and transformed to chloride form with 10 mL of concentrated HCl and 3–4 drops of H_2_O_2_. The samples were evaporated to dryness again and prepared in this way for autodeposition on silver. The silver was cut into 1 cm discs and then washed with plenty of acetone to get rid of any residual fat. The autodeposition process took place after the dissolution of the sample in 10 mL of 0.5 M HCl, and about 0.05 g of ascorbic acid was added to keep the iron in oxidation state II. The sample was quantitatively transferred into a PTFE cell, on the bottom of which there was a silver disc. The vessel was placed in a 90 °C water bath under constant mixing for 4 h in order to deposit more than 99% of polonium [53]. Thereafter, the plate was removed, washed with acetone and water and deposited for measurement in an alpha spectrometer.

The activity concentrations of ^210^Po in the wine samples were measured using an alpha spectrometer (Alpha Analyst S470, Oak Ridge, TN, USA). The device was equipped with a PIPs detector with an active surface of 300–450 mm^2^. The detector was connected to a 1024 multichannel analyzer (Canberra–Packard, Parkway Meridian, Meriden, CT, USA) and placed in a vacuum chamber. The detector efficiency ranged from 0.33 to 0.35, and the resolution ranged from 17–18 keV. The minimum detectable activity was 0.003 mBq/g (bottom sediments). The accuracy was set to less than 7%. The method was verified on the basis of appropriate interlaboratory tests, such as international comparative testing and the analysis of IAEA materials (IAEA-327, 384, 385, 414, IAEA-TEL-2011-03 and MODAS-2015). The reference sample matrices were fish tissues, bottom sediments and acidified waters. On their basis, the parameters and the efficiency of the method were measured. Polonium activity was corrected by the time from being deposited to measurement. The results of measurements of the activity concentrations of ^210^Po were expressed per 1 L of wine volume.

### 3.3. Evaluation of Annual Effective Dose Due to the ^210^Po

In order to estimate the exposure of adult Polish consumers of fruit wines to ionizing radiation, the annual effective dose for ^210^Po was computed from the following equation:D^210^Po = Q × C^210^Po × I(1)
where D_210Po_—annual effective dose (nSv/y); Q—the dose conversion factor (1.2 × 10^−6^ Sv/Bq); C_210Po_—activity concentration (Bq/L); and I—average wine consumption. The dose conversion factor for ^210^Po for adults was provided by the ICRP is 1.2 (µSv.Bq^−1^) [54]. An average intake of 9.5 L per adult was used to represent the annual wine consumption in Poland.

### 3.4. Data Analysis

All analyses were performed in triplicate and potential measurement outliers were searched. As the data for every voivodeship were strongly skewed, they were transformed to logarithms, and then we computed the means and SD values and performed an analysis of variance to evaluate the between-region differences, whereby the level *p* ≤ 0.05 was considered significant.

## 4. Conclusions

The ^210^Po activity concentration in fruit wines may be arbitrarily classified into four groups: very low (<10 mBq/L);low (10–100 mBq/L);medium (101–200 mBq/L);high (>200 mBq/L).

The wide range of the activity concentration of ^210^Po in fruit wines resulted from different environmental conditions in the studied areas, e.g., soil parameters (uranium concentration) and the sum of the precipitation and terrain characteristics, e.g., the proximity of the lakes. Strong pollution of the environment by aqueous wastes related to coal energy production, coal fly ash and flue gases and particulate air pollution in the Małopolska voivodeship probably contributed to a much higher activity concentration of ^210^Po in wine from that site as compared with the other locations.

The estimated levels of the annual effective dose due to the ingestion of fruit wines varied from 0.312 to 1.214 µSv/year. These levels of exposure are relatively safe compared with the WHO levels for specific radionuclides, having a generic criterion of 0.1 mSv per year for ingestion, and with the IAEA reference level for public exposure in specific existing exposure situations including food, whose criterion is 1 mSv per year, which was not exceeded in any sample. However, the long-term effects of low doses of polonium exposure are unknown. Thus, there is a need to conduct new research using modern laboratory tools.

As the samples were collected from a relatively small area of Poland in this study, it was not possible to make a recommendation of a suitable location for producing fruit wines. Further work needs to be undertaken to examine differentiation in ^210^Po activity concentrations in samples derived from all vineyard regions in Poland to provide greater confidence in assigning the location for wine production and for monitoring radionuclide activity.

## Figures and Tables

**Figure 1 molecules-28-00438-f001:**
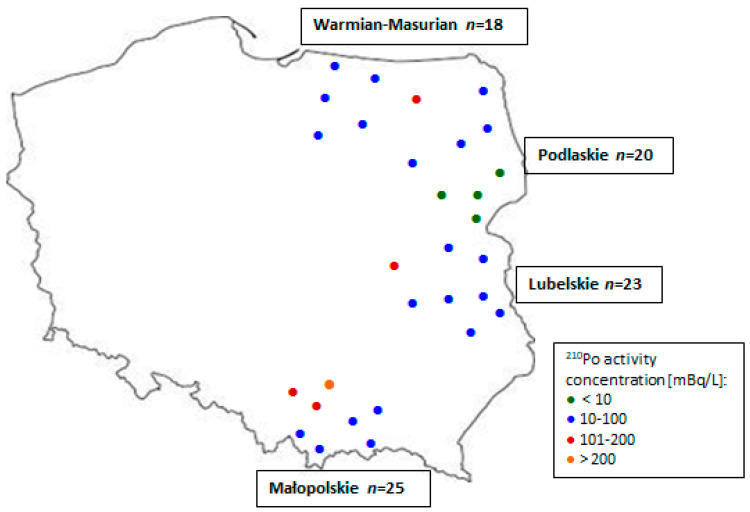
Categories of ^210^Po activity concentration (mBq/L) in fruit wine samples derived from different locations in Poland.

**Table 1 molecules-28-00438-t001:** ^210^Po activity concentration (mBq/L) in fruit wine samples derived from different locations in Poland.

Production Site	Mean, SD	Range
Warmian–Masurian	42.24 × 1.85 ^±1,a^	18.6–125.3
Podlaskie	16.58 × 2.83 ^±1,b^	3.5–97.1
Małopolskie	98.67 × 1.46 ^±1,c^	61.4–221.4
Lubelskie	71.85 × 1.43 ^±1,c^	38.7–169.1

^a,b,c^—values bearing different superscripts in column differ significantly from each other, *p* ≤ 0.05.

**Table 2 molecules-28-00438-t002:** Characteristics of production location of studied fruit wines.

Variable	Site			
	Warmian–Masurian	Podlaskie	Lubelskie	Małopolskie
Concentration of uranium in soil (mg/kg) [40]	0.5–1.6	0.5–1.6	0.5–1.6	2.5–3.4
Total annual precipitation (mm)	684	585	560	780
PM_2.5_ (μg/m^3^)	15.0	15.8	19.0	29.0
PM_10_ (μg/m^3^)	18.1	19.8	21.0	30.7

PM—particulate matter with a diameter less than 2.5 (PM_2.5_) and 10 μm (PM_10_).

**Table 3 molecules-28-00438-t003:** Annual effective dose (μSv/year) for ^210^Po due to the ingestion of fruit wines from different locations.

Production Site	Mean, SD	Range
Warmian–Masurian	0.482 × 1.853 ^±1,a^	0.112–1.430
Podlaskie	0.191 × 2.811 ^±1,b^	0.041–1.108
Małopolskie	1.126 × 1.464 ^±1,c^	0.700–2.572
Lubelskie	0.820 × 1.434 ^±1,c^	0.443–1.930

^a,b,c^—values bearing different superscripts in column differ significantly from each other, *p* ≤ 0.05.

## Data Availability

Data available from the corresponding authors upon reasonable request.

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
