# Peer review of "Study of Polonium (210Po) Activity Concentration in Fruit Wines Derived from Different Locations in Poland"

_molecules, 2023, doi:10.3390/molecules28010438_

Round 1

Reviewer 1 Report (Previous Reviewer 2)

The manuscript has been revised as advised by the reviewer. 

Reviewer 2 Report (New Reviewer)

The revised manuscript adequately deals with the important study of polonium (210Po) activity concentration in fruit wines, the origin of which are various locations in Poland, which are smaller in area and cannot be generalized to all areas, including recommendations.

The problem is very well and comprehensively presented in the introduction, from natural radiation, toxicity and absorption, contamination of various foods.

A total of 86 samples of fruit wines from smaller producers were analysed. The samples represent different fruit types, colours and sugar levels of fruit wines, which also gives the original article practical value. The results are presented in three tables and one figure, and the discussion is sufficiently comprehensive and supported by relevant literature sources. The authors cite a total of 55 current and relevant sources.

The conclusions of the study summarize that the data obtained provide information on the activity concentration of 210Po of fruit wines and in crease databases on natural radioactivity of foodstuffs. Future work is needed to examine 210Po activity in samples from all vineyard regions in Poland.

I have no comments or corrections, so I am not attaching the manuscript either.

This manuscript is a resubmission of an earlier submission. The following is a list of the peer review reports and author responses from that submission.

Round 1

Reviewer 1 Report

General comments:

A very strange paper indeed and certainly not one worthy of publication.  The authors appear to have decided that there is polonium in wine (I don’t dispute that albeit at minuscule quantities) but also that it’s dangerous, irrespective of the outcome of their analysis, see for example last line in the abstract.  I do dispute this because the levels reported do not support this view i.e. 'should also be considered to ensure consumer safety'.  No hypothesis or justification for the work is provided and, indeed, the prior Italian data stated early in the work suggests that, actually, there isn’t a case for the work based on the very low activities measured.  This is reinforced by the statement later in the work ‘The effective doses due to the ingestion of radionuclides are harmless only when they do not exceed the 1 mSv limit’, bearing in mind that this is wrong in any case, i.e., stochastic harm can result at low doses based on the linear, no-threshold principle.  The work is also out of scope for the Molecules journal and, were it of a publishable standard, it would be better suited to a journal specialising in environmental radioactivity.  However, I return to the problem with the absent hypothesis: what is the justification for looking at: i) wine, ii) these wines in particular, and iii) the 210Po isotope and so forth?  This is not given.

It is necessary to revise two important, fundamental basic tenets of quantitative scientific analysis for the review of this work: firstly, and on a specific issue, annual background dose in Poland is, say, ~2.48 mSv per year, so anything worthy of concern needs to exceed this (and w.r.t. 210Po in wine it doesn’t); secondly elementary statistical basis for assessing consistency is that 99.7% of the data will fall within 3 standard deviations of the mean, assuming the data are Gaussian distributed, which would appear reasonable in this case.  However, none of the data provided fall into this category despite that suggestion in the paper that there are statistically justified departures from consistency that warrant further interpretation; they don’t.  In summary, very small quantities of 210Po are observed (ok) that are consistent with take-up by the vines from the decay of trace quantities of naturally occurring uranium in the ground, and some variation is observed as one would expect given the different locations in which the vines are grown.  None of this is a surprise or remarkable.

Detailed comments:

Line 13: polonium-210

15: what is the ‘radiological hazard effect’?  Explain.

21: the data do not appear significantly different from one another but, rather, they are entirely consistent on a statistical basis: – 48+/- 16 and 61+/ 20

23: ingestion, but how much?  This is important in terms of estimating internal dose.

26: last line, really?  Their specific radiotoxicity is way lower than 210Po, and on what basis – are you implying the risk ranks for example with cirrhosis of the liver from excess alcohol consumption, that I'd argue even then would not even approach a degree of radiological exposure that warrants further consideration in terms of health detriment?

36: ‘the’ winemaking process

41: I don’t understand: 210Po does not have a small mass.  You mean I think that it has a high ‘specific activity’.

44: as above, yes 210Po does have a high radiotoxicity but this does not mean it is the most dangerous, particularly because it has to be ingested due to its alpha-based mode of decay.  There are other ‘dangerous’ isotopes, in terms of being a ‘hazard’, consider for example the plutonium isotopes.  Further, 210Po generally has to be made artificially in quantities to render it a plausible concern which is not easy.

59: but, basically, the 210Po you are concerned with all comes from decay of natural uranium in the ground (?) in quantities that are very small even in uranium-rich ores and which one would not normally associate with the sites from which the wines studied derive in this work, in any case.

65: there doesn’t appear to be any justification for this statement, why should it depend on grape variety?

66 et seq: It is a nonsense to quote uncertainties to as much accuracy as the datum itself, and 1 significant figure in an uncertainty will usually suffice (otherwise it is implied that the uncertainty is known to a greater degree of accuracy than the datum, which cannot be right).  Hence, these data should be stated: 4+/-2 mBq/l, 4+/-3 mBq/l etc. etc. and, consequently, because the uncertainties are large therefore the potential for insight is low.

70: 0.5 litre per day for a lifetime will cause other significant health problems especially cirrhosis etc. likely on a deterministic basis that far outweigh anything radiological, i.e., 0.001 mSv/year is less than a thousand of natural background – so this work is irrelevant in terms of risk to health, is it not?  Also, what is the evidence that the entire inventory of 210Po in a 0.5 litre exposes internal tissues equivalently!?

98 et seq.: yes, well it will be different depending on region (see above in general comments) because you’re seeing the variation in natural mineral composition (uranium) from place to place in the Earth's crust which certainly does not have to be consistent but, as per Fig. 1, the data from different places do appear consistent with one another, bu with the Warmian samples showing less confidence as a measurement, but I’d say that’s the limit of the interpretation that’s possible.

111: are you sure?  Where do you think the 210Po is coming from in the atmosphere? It can’t be fallout as its half-life is too short, given that there have been no bomb tests for many years, similarly for reactor accidents such as Chernobyl and Fukushima.  I suspect the 210Po is coming from uranium in the ground with a smaller amount of weather-related and localised ‘lift’ that might settle on leaves, grapes etc.

Fig 3 and 4 – all data reads as consistent with itself, implying no variation and the data do not ‘differ significantly’.  Ditto Figure 5.

At this stage in the paper, it transpires that (line 177) that hardly any wine is drunk in Poland in any case, despite the earlier estimate of 0.5 litres per day. Hence, why is this work being done?

186: ‘The effective doses due to the ingestion of radionuclides are harmless only when 186 they do not exceed the 1 mSv limit.’  This statement sums up the significant shortcomings of this paper: the authors do not appear to understand the fundamental basis of radiation protection and health physics – that is, the distinction of deterministic and stochastic health effects, and the need to minimise exposure 'as low as reasonably practicable, ALARP'.  At the levels presented, we are faced with a consideration of stochastic health effects, i.e., predominantly cancer, and hence the ALARP philosophy .  Doses < 1 mSv cannot be considered as 'harmless', per se, but certainly at small doses it is very difficult to discern a causal link with exposure and effects that might follow, often a very long time afterwards and which are likely to be unrelated to the exposure.  Rather, dose limits are set to prevent the habitual activity of a person at risk in a controlled environment from ever accruing a dose that might pose a significant risk and, indeed, if such a limit was exceeded an investigation would result.  Of course, this dose limit in the context of this work is irrelevant any way because there is not sufficient 210Po and not enough wine is being consumed.

Author Response

Dear Reviewer,

        We are greatly obliged for having received your opinions, constructive comments, and helpful suggestions on our manuscript. These comments are valuable and very helpful to revise and improve our manuscript. The replies to point-to-point to the specific comments are listed below. All changes in the manuscript are marked in red.

General comments

Regarding the Reviewer’s general comment 1: “A very strange paper indeed and certainly not one worthy of publication. The authors appear to have decided that there is polonium in wine (I don’t dispute that albeit at minuscule quantities) but also that it’s dangerous, irrespective of the outcome of their analysis, see for example last line in the abstract. I do dispute this because the levels reported do not support this view i.e. 'should also be considered to ensure consumer safety'.”

Accordingly linear no-threshold hypothesis, radiation doses greater than zero will increase the risk of excess cancer or heritable disease in a simple proportionate manner in the low-dose range. Linear no-threshold model (LNT) is a dose-response model used in radiation protection to estimate stochastic health effects such as radiation-induced cancer, genetic mutations and teratogenic effects on the human body due to exposure to ionizing radiation. The LNT model is commonly used by regulatory bodies as a basis for formulating public health policies that set regulatory dose limits to protect against the effects of radiation.

The small dose is not necessarily dangerous to health, but undoubtedly unfavorable. However, it would be important in the context of regular and long-term wine consumption. Topic of the manuscript is important, because of dynamic development of wine production in Poland and expected increase of wine consumption.

Regarding the Reviewer’s general comment 2: “No hypothesis or justification for the work is provided and, indeed, the prior Italian data stated early in the work suggests that, actually, there isn’t a case for the work based on the very low activities measured.  This is reinforced by the statement later in the work ‘The effective doses due to the ingestion of radionuclides are harmless only when they do not exceed the 1 mSv limit’, bearing in mind that this is wrong in any case, i.e., stochastic harm can result at low doses based on the linear, no-threshold principle.”

 The hypothesis was determined in terms of the radiation dose. The study aimed to determine the possible hazard of wine consumption in Poland, including each type of wine.

Regarding the Reviewer’s general comment 3: The work is also out of scope for the Molecules journal and, were it of a publishable standard, it would be better suited to a journal specialising in environmental radioactivity.”

We would like to explain that the topic of the paper fits the scope of the special issue of the Molecules journal: “Environment radioactivity analysis, monitoring and tracer applications”. An invitation to submit manuscripts was addressed to Researchers working in the environmental radioactivity field and related fields, associated with all aspects of method development for environmental radioassays, observations in environmental radioactivity monitoring and the environmental applications of radionuclides as tracers.

Regarding the Reviewer’s general comment 4: “However, I return to the problem with the absent hypothesis: what is the justification for looking at: i) wine, ii) these wines in particular, and iii) the 210Po isotope and so forth? This is not given.”

We would like to thank you for this comment. We added information in Introduction section accordingly Reviewer’s suggestion.

Regarding the Reviewer’s general comment 5:It is necessary to revise two important, fundamental basic tenets of quantitative scientific analysis for the review of this work: firstly, and on a specific issue, annual background dose in Poland is, say, ~2.48 mSv per year, so anything worthy of concern needs to exceed this (and w.r.t. 210Po in wine it doesn’t).”

We would like to explain that we not found such information about annual background dose in Poland ~ 2.48 mSv per year (in published papers and any sources).

The last research BoryÅ‚o et al. 2022 (published in International Journal Environmental Research Public Health) showed that an adult Polish resident receives about annual effective dose of 309 μSv of 210Po and 210Pb from inhalation and ingestion. Smoking cigarettes and marijuana revealed as main sources of both radionuclides. Consumption of wines were not included in computing this dose (in this paper).

Regarding the Reviewer’s general comment 6: “Secondly elementary statistical basis for assessing consistency is that 99.7% of the data will fall within 3 standard deviations of the mean, assuming the data are Gaussian distributed, which would appear reasonable in this case. However, none of the data provided fall into this category despite that suggestion in the paper that there are statistically justified departures from consistency that warrant further interpretation; they don’t.”

We would like to explain that the chi squared test did not confirm normal distribution of analytical data (this information was inserted in 3.4. part: Statistical analysis. Thus, the Kruskal-Wallis one-way analysis of variance test was applied to assay possible differences in samples from two locations.

Regarding the Reviewer’s general comment 7: In summary, very small quantities of 210Po are observed (ok) that are consistent with take-up by the vines from the decay of trace quantities of naturally occurring uranium in the ground, and some variation is observed as one would expect given the different locations in which the vines are grown. None of this is a surprise or remarkable.”

Yes, we agree that Polonium-210 (210Po), an a-emitting naturally occurring radionuclide exists in the environment as a result of Uranium-238 (238U) decay chain. This information is included in the second paragraph of Introduction part. However, we also hypothesize that 238U decay chain were not only one source of 210Po in plants (accordingly results of some studies).

In our study the locations were selected for sampling because soils in the north-eastern region of Poland contain low amounts of Uranium (the lowest in Europe) and this region is not industrialized.

Detailed comments:

Regarding the Reviewer’s detailed comment 1: “Line 13: polonium-210”

It was corrected accordingly Reviewer’s suggestion.

Regarding the Reviewer’s detailed comment 2: 15: what is the ‘radiological hazard effect’? Explain”.

We are sorry for mistake; the sentence was corrected.

Regarding the Reviewer’s detailed comment 3: 21: “the data do not appear significantly different from one another but, rather, they are entirely consistent on a statistical basis: – 48+/- 16 and 61+/20”.

We would like to explain that statistical analysis (using Kruskal-Wallis one-way analysis of variance test) revealed significant differences in data of wines derived from two locations (Warmian-Masurian and Podlaskie). Such significant differences in data obtained from wines of two locations is shown on Figure 1 in the manuscript.

Regarding the Reviewer’s detailed comment 4: 23: “ingestion, but how much? This is important in terms of estimating internal dose.”

The sentence was revised and supplemented accordingly Reviewer’s suggestion.

Regarding the Reviewer’s detailed comment 5: 26: “last line, really?  Their specific radiotoxicity is way lower than 210Po, and on what basis - are you implying the risk ranks for example with cirrhosis of the liver from excess alcohol consumption, that I'd argue even then would not even approach a degree of radiological exposure that warrants further consideration in terms of health detriment?”

We would like to explain that the study aimed to determine the activity concentration of polonium-210 in Polish wines and to assess the exposure of the adults to ionizing radiation from 210Po in investigated wines in terms of their annual effective dose. The aim of the study was not to investigate the harmful effects of alcohol consumption on human health. In cited by us study (Desideri et al. 2010). Although annual consumption of wine in Italy obtained 46.6 L, in study Authors declared 0.5 L daily consumption of wine.

Regarding the Reviewer’s detailed comment 6: 36: ‘the’ winemaking process”

It was corrected accordingly Reviewer’s suggestion.

Regarding the Reviewer’s detailed comment 7: 41: “I don’t understand: 210Po does not have a small mass. You mean I think that it has a high ‘specific activity’.”

Thank you for that comment. We agree. The sentence was revised, and Introduction apart was supplemented with appropriate information from literature data.

Regarding the Reviewer’s detailed comment 8: “44: as above, yes 210Po does have a high radiotoxicity but this does not mean it is the most dangerous, particularly because it has to be ingested due to its alpha-based mode of decay. There are other ‘dangerous’ isotopes, in terms of being a ‘hazard’, consider for example the plutonium isotopes. Further, 210Po generally has to be made artificially in quantities to render it a plausible concern which is not easy.”

210Po represents a high radiotoxicity. Studies of Jefferson et al. 2009 and Harrison et al. 2007 revealed that Po-210 is 20 times more toxic than hydrogen cyanide, hence it ca be carcinogenic to humans if ingested. Introduction part was also supplemented with information about fatal acute dose for humans.

Ingested 210Po is more readily absorbed to blood than some other alpha-emitting radionuclides, e.g. plutonium-239.

Harrison, J.; Leggett, R.; Lloyd, D.; Phipps, A.; Scott, B. Polonium-210 as a poison, J. Radiol. Prot. 2007, 27, 17-40.

Jefferson, R.D.;  Goans, R.E.; Blain P.G.; Thomas, S.H.L. Diagnosis and treatment of polonium poisoning. Clin. Toxicol. 2009, 47, 379-392.

Regarding the Reviewer’s detailed comment 9: 59: “but, basically, the 210Po you are concerned with all comes from decay of natural uranium in the ground (?) in quantities that are very small even in uranium-rich ores and which one would not normally associate with the sites from which the wines studied derive in this work, in any case.”

The regions of Poland were selected because of the lowest level of uranium concentration in soil in Europe. In addition, these are low-urbanized areas and have not large industrial facilities that could disturb the measurements. This has proved to be valuable in assessing the differences between different types of wine (e.g. for the difference between red and white). Italy does not have any region highly contaminated with uranium, therefore the uranium doses in the publication are very low and the measurements in some cases have even 50% error. Hence, we decided that uranium will not be significant due to the very low level of contamination with this element in Poland. Mino higher uranium contamination on the Apennine Peninsula does not show any major differences compared to Poland in terms of polonium concentrations. Relevant fragment has been added.

Desideri, D.; Roselli, C.; Meli, A. Intake of 210Po, 234U and 238U radionuclides with wine in Italy, Food Chem Toxicol. 2010, 48, 650-657. DOI: 10.1016/j.fct.2009.11.047

https://remon.jrc.ec.europa.eu/About/Atlas-of-Natural-Radiation/Digital-Atlas/Uranium-in-soil/Uranium-concentration-in-soil-

Regarding the Reviewer’s detailed comment 10: 65: “there doesn’t appear to be any justification for this statement, why should it depend on grape variety?”

It has been proved in the publication that the color of the wine does not really matter. But it is also valuable information pointing to a consistent dose from both types of wine. Thanks to this, it is possible to unify the method of calculating the dose without dividing it into wine color.

Regarding the Reviewer’s detailed comment 11: 66 et seq: It is a nonsense to quote uncertainties to as much accuracy as the datum itself, and 1 significant figure in an uncertainty will usually suffice (otherwise it is implied that the uncertainty is known to a greater degree of accuracy than the datum, which cannot be right).  Hence, these data should be stated: 4+/-2 mBq/l, 4+/-3 mBq/l etc. etc. and, consequently, because the uncertainties are large therefore the potential for insight is low.”

We are sorry for that presentation of data which we inserted directly from the cited paper. We are grateful for helpful suggestion. It was corrected accordingly Reviewer’s suggestion.

Regarding the Reviewer’s detailed comment 12: 70: “0.5 litre per day for a lifetime will cause other significant health problems especially cirrhosis etc. likely on a deterministic basis that far outweigh anything radiological, i.e., 0.001 mSv/year is less than a thousand of natural background - so this work is irrelevant in terms of risk to health, is it not? Also, what is the evidence that the entire inventory of 210Po in a 0.5 litre exposes internal tissues equivalently!?”

Wine consumption in Poland is much lower than in Italy, which was included in the study during the calculations. (Data are provided under the calculation formula) The authors do not comment on alcohol consumption in other countries. This is not the purpose of the work. The dose is determined and calculated from the data. These doses are part of the total dose in Poland for an adult.

Regarding the Reviewer’s detailed comment 13: “98 et seq.: yes, well it will be different depending on region (see above in general comments) because you’re seeing the variation in natural mineral composition (uranium) from place to place in the Earth's crust which certainly does not have to be consistent but, as per Fig. 1, the data from different places do appear consistent with one another, but with the Warmian samples showing less confidence as a measurement, but I’d say that’s the limit of the interpretation that’s possible.”

The regions of Poland were selected because of the lowest level of uranium concentration in soil in Europe. In addition, these are low-urbanized areas and do have not large industrial facilities that could disturb the measurements. Another factor is that these doses can be applied to the entire territory of Poland due to the low variation in the concentrations of radioactive elements in the soil in Poland.

Desideri, D.; Roselli, C.; Meli, A. Intake of 210Po, 234U and 238U radionuclides with wine in Italy, Food Chem Toxicol. 2010, 48, 650-657. DOI: 10.1016/j.fct.2009.11.047

https://remon.jrc.ec.europa.eu/About/Atlas-of-Natural-Radiation/Digital-Atlas/Uranium-in-soil/Uranium-concentration-in-soil-

Regarding the Reviewer’s detailed comment 14: 111: are you sure?  Where do you think the 210Po is coming from in the atmosphere? It can’t be fallout as its half-life is too short, given that there have been no bomb tests for many years, similarly for reactor accidents such as Chernobyl and Fukushima.  I suspect the 210Po is coming from uranium in the ground with a smaller amount of weather-related and localised ‘lift’ that might settle on leaves, grapes etc.”

As many publications indicate, one of the main sources of polonium in plants is wet and dry deposition. Extraction of uranium and polonium from soil in plants is negligible. Both in plants of the same order, such as hemp or nettles, and in other plants, such as tobacco.

G. Olszewski, A. BoryÅ‚o, B. Skwarzec: A study on possible use of Urtica dioica (common nettle) plant as polonium 210Po and lead 210Pb contamination biomonitor in the area of phosphogypsum stockpile. Environmetal Sciences and Pollution Research, 2016, 23, 6700-6708.

G. Olszewski, A. BoryÅ‚o, B. Skwarzec: A study on possible use of Urtica dioica (common nettle) plants as uranium (234U, 238U) contamination bioindicator near phosphogypsum stockpile. Journal of Radioanalytical and Nuclear Chemistry, 2016, 308, 37-46.

J. Wieczorek, M. Kaczor, A. BoryÅ‚o: Determination of 210Po and 210Pb in cannabis (Cannabis sativa L.) plants and products. Journal of Environmental Radioactivity, 2022, 246, 106834.

B. Skwarzec, D. Strumińska, J. Ulatowski, M. Kwiatkowski: Determination and Distribution of 210Po in Tobacco Plants from Poland. Journal of Radioanalytical and Nuclear Chemistry, 2001, 250, 319-322.

Regarding the Reviewer’s detailed comment 15: Fig 3 and 4 - all data reads as consistent with itself, implying no variation and the data do not ‘differ significantly’. Ditto Figure 5.

Yes, all data read as consistent because applied statistical test do not show significant differences between variables. So, differences in activity concentrations of 210Po depending on wine color or type of fruits used for wine productions are not relevant from the statistical point of view.

Regarding the Reviewer’s detailed comment 16: At this stage in the paper, it transpires that (line 177) that hardly any wine is drunk in Poland in any case, despite the earlier estimate of 0.5 litres per day. Hence, why is this work being done?

Wine consumption in Poland is much lower than in Italy, which was included in the study during the calculations. (Data are provided under the calculation formula) The authors do not comment on alcohol consumption in other countries. This is not the purpose of the work. The dose is determined and calculated from the data. These doses are part of the total dose in Poland for an adult.

Regarding the Reviewer’s detailed comment 17: 186: ‘The effective doses due to the ingestion of radionuclides are harmless only when they do not exceed the 1 mSv limit.’  This statement sums up the significant shortcomings of this paper: the authors do not appear to understand the fundamental basis of radiation protection and health physics – that is, the distinction of deterministic and stochastic health effects, and the need to minimise exposure 'as low as reasonably practicable, ALARP'.  At the levels presented, we are faced with a consideration of stochastic health effects, i.e., predominantly cancer, and hence the ALARP philosophy. Doses < 1 mSv cannot be considered as 'harmless', per se, but certainly at small doses it is very difficult to discern a causal link with exposure and effects that might follow, often a very long time afterwards and which are likely to be unrelated to the exposure.  Rather, dose limits are set to prevent the habitual activity of a person at risk in a controlled environment from ever accruing a dose that might pose a significant risk and, indeed, if such a limit was exceeded an investigation would result.  Of course, this dose limit in the context of this work is irrelevant any way because there is not sufficient 210Po and not enough wine is being consumed.

Direct effects of radiation doses such as cancer or any kind of mutation are extremely rounds to be proven especially for low doses. But environmental control of food safety should be possible broad and precise to align the amount of the dose limit with the homeostasis of human health. According to ICRP 2012, the value of the effective dose should not exceed 1mSv. Doses in the order of 2.48 mSv can already pose a high health risk. Each additional information on the potential radiation dose in the human environment is valuable and will bring added value. According to the linear hypothesis, even the lowest dose of ionizing radiation has a negative impact on health. This is in line with the ALARP principle. Taking into account other sources of radiation, the dose exceedance is not completely ruled out.

Reviewer 2 Report

The manuscript is well written however, following concerns should be addressed.

Introduction: Provide comprehensive background on harmful effect or hazards associated with 210Po and what are maximum allowable limits if regulations defines, in international and local perspective. Also enlist all major factors contributing its existence in wines.

Results:

More justification is required to support why two different sites had significantly different levels of 210Po

All bar graphs should be incorporated with statistical significance symbols. 

Similarly extensive justification is required for variations in Other European wines and Poland's wines. What are the possible reasons and how can be overcome. 

Table 1 and 2 are two small, rather they can be described as text. as just 2 row table doses not present significant comparison.

Figure 7. A clear and confined sampling locations can be accessed freely with marked sites on map. In current form it is meaningless.

The authors need to include extensive discussion to support their findings. How this problem can be overcomes and what could be the long term consequences.

Author Response

Dear Reviewer,

        We are greatly obliged for having received your opinions, constructive comments, and helpful suggestions on our manuscript. These comments are valuable and very helpful to revise and improve our manuscript. The replies to point-to-point to the specific comments are listed below. All changes in the manuscript are marked in red.

Regarding the Reviewer’s general comment:

The manuscript is well written however, following concerns should be addressed.

Regarding the Reviewer’s comment about Introduction part: “Introduction: Provide comprehensive background on harmful effect or hazards associated with 210Po and what are maximum allowable limits if regulations defines, in international and local perspective. Also enlist all major factors contributing its existence in wines.”

Information was added accordingly Reviewer’s suggestion.

Regarding the Reviewer’s comments about Introduction about Results part:

Comment 1: More justification is required to support why two different sites had significantly different levels of 210Po

The study covers the areas of northern Poland due to the low diversity of this area in the context of the content of radioactive elements in the soil. Especially uranium in the soil. An additional factor is the low level of urbanization of the area and low-developed light and heavy industry. It is also possible to consider extrapolating the results to the entire territory of Poland due to the low geological diversity of soils. As shown by the research of honey and milk on the content of 210Po carried out in previous years, the high level of uranium in the soil and the industrialization of the area may have an impact on the isotopic composition of food. The authors wanted to avoid additional sources of matrix polonium contamination.

Comment 2: All bar graphs should be incorporated with statistical significance symbols.

The significance level was a = 0.05 in each case. This information was added under each figure.  

Comment 3: Similarly extensive justification is required for variations in Other European wines and Poland's wines. What are the possible reasons and how can be overcome. 

It is very difficult to refer to publications on the content of radioisotopes in wines due to the innovation of research. The only reference point is the Italian publication, and the authors refer to it extremely often. Due to the similar matrix, the tests can be compared to water, beer, and milk. Milk shows a much greater possibility of accumulation and biomagnification of polonium, hence the higher dose. Water shows a similar activity to wine. The concentration of 210Po in beer also remained at a similar level.

Desideri, D.; Roselli, C.; Meli, A. Intake of 210Po, 234U and 238U radionuclides with wine in Italy, Food Chem Toxicol. 2010, 48, 650-657. DOI: 10.1016/j.fct.2009.11.047

Boryło, A., Skwarzec, B.; Wieczorek, J. Sources of Polonium 210Po and radio-lead 210Pb in human body in Poland. Int J Environ Res Public Health. 2022, 19, 1984. doi.org/10.3390/ijerph19041984

Skwarzec, B.; Strumińska, D.I.; Boryło, A.; Falandysz, J. Intake of 210Po, 234U and 238U radionuclides with Beer in Poland. Journal of Radioanalytical and Nuclear Chemistry, 2004, 261, 661-663.

BoryÅ‚o, A.; Kaczor, M.; Wieczorek, J.; RomaÅ„czyk, G. Assessment of intake 210Po and 210Pb isotopes from cow’s milk in Poland. Isotopes in Environmental and Health Studies, 2021, 57, 623-631.

Comment 4: Table 1 and 2 are two small, rather they can be described as text. as just 2 row table doses not present significant comparison.

Tables 1 and 2 were removed from manuscript accordingly Reviewer’s suggestion.

Comment 5: Figure 7. A clear and confined sampling locations can be accessed freely with marked sites on map. In current form it is meaningless.

Locations were added on map (Figure 7) accordingly Reviewer’s suggestion.

Comment 6: The authors need to include extensive discussion to support their findings. How this problem can be overcomes and what could be the long-term consequences.

It is very difficult to refer to publications on the content of radioisotopes in wines due to the innovation of research. The only reference point is the Italian publication, and the authors refer to it extremely often. Due to the similarity matrix, the tests can be compared to water, beer, and milk. Milk shows a much greater possibility of accumulation and biomagnification of polonium, hence the higher dose. Water shows similar activity to wine. The concentration of 210Po in beer also remained at a similar level.

Boryło, A., Skwarzec, B.; Wieczorek, J. Sources of Polonium 210Po and radio-lead 210Pb in human body in Poland. Int J Environ Res Public Health. 2022, 19, 1984. doi.org/10.3390/ijerph19041984

Skwarzec, B.; Strumińska, D.I.; Boryło, A.; Falandysz, J. Intake of 210Po, 234U and 238U radionuclides with Beer in Poland. Journal of Radioanalytical and Nuclear Chemistry, 2004, 261, 661-663.

BoryÅ‚o, A.; Kaczor, M.; Wieczorek, J.; RomaÅ„czyk, G. Assessment of intake 210Po and 210Pb isotopes from cow’s milk in Poland. Isotopes in Environmental and Health Studies, 2021, 57, 623-631.

Reviewer 3 Report

The manuscript entitled “Study of polonium (210Po) activity concentrations in wines from north-eastern region of Poland” reports the activity of 210Po in wines from two different geographic region of Poland and from European. In my opinion, this manuscript should be submitted to major revisions before publication since some conclusions are difficult to understand. As an example, Figure 2, activity concentration of 210Po in polish wines depending on wine color? How is possible? Or a possible explanation is the winemaking used to obtain this wine or the raw material used?

Figure 3. activity concentration of 210Po in polish wines depending on type of fruits?? These fruits were collected in the same geographical region.

In addition, the results section should be rewritten and compared with previous studies.

The figures resolution should be improved.

Line 98: at the 0.05 significance level, the p value was 0.006??? What does this mean?

Author Response

Dear Reviewer,

        We are greatly obliged for having received your opinions, constructive comments, and helpful suggestions on our manuscript. These comments are valuable and very helpful to revise and improve our manuscript. The replies to point-to-point to the specific comments are listed below. All changes in the manuscript are marked in red.

Regarding the Reviewer’s general comment 1: “Figure 2, activity concentration of 210Po in polish wines depending on wine color? How is possible? Or a possible explanation is the winemaking used to obtain this wine or the raw material used ?”Regarding the Reviewer’s general comment 2: “Figure 3. activity concentration of 210Po in polish wines depending on type of fruits?? These fruits were collected in the same geographical region.”

One of the research hypotheses was an attempt to prove possible differences in the concentrations of polonium in wines of different types (red and white), which was aimed at clarifying the method of calculating the dose. The hypothesis was made because of the morphological differences of the various fruits from which the wine is produced. During the research we find that the dose can be calculated in the same way, regardless of the type and taste of the wine. We prove it with appropriate statistical tests. All the admixed fruits and the grapes themselves were harvested only at the production sites.

Regarding the Reviewer’s general comment 3:In addition, the results section should be rewritten and compared with previous studies.”

Due to the similar matrix, the tests can be compared to water, beer, and milk. Milk shows a much greater possibility of accumulation and biomagnification of polonium, hence the higher dose. Water shows a similar activity to wine. The concentration of 210Po in beer also remained at a similar level. This informations were also inserted in conclusions section.

  1. Boryło, A., Skwarzec, B.; Wieczorek, J. Sources of Polonium 210Po and radio-lead 210Pb in human body in Poland. Int J Environ Res Public Health. 2022, 19, 1984. org/10.3390/ijerph19041984
  2. Skwarzec, B.; Strumińska, D.I.; Boryło, A.; Falandysz, J. Intake of 210Po, 234U and 238U radionuclides with Beer in Poland. Journal of Radioanalytical and Nuclear Chemistry, 2004, 261, 661-663.
  3. BoryÅ‚o, A.; Kaczor, M.; Wieczorek, J.; RomaÅ„czyk, G. Assessment of intake 210Po and 210Pb isotopes from cow’s milk in Poland. Isotopes in Environmental and Health Studies, 2021, 57, 623-631.

Regarding the Reviewer’s general comment 4:The figures resolution should be improved.”

Figure 6 was corrected accordingly Reviewer’s suggestion.

Regarding the Reviewer’s general comment 5:Line 98: at the 0.05 significance level, the p value was 0.006??? What does this mean?”

A p-value measures the probability of obtaining the observed results, assuming that the null hypothesis (there are no statistically significant differences between values or data) is true. The lower the p-value, the greater the statistically significance of the observed difference. A p-value of 0.05 or lower is generally considered statistically significant.  

Round 2

Reviewer 2 Report

The manuscript has been revised, but only concern here is that the comparison studies and level of detection and method validation such HPLC method develop details should be mentioned in supplementary materials.

Reviewer 3 Report

The manuscript entitled “Study of polonium (210Po) activity concentrations in wines from north-eastern region of Poland” should be accepted in the current form, since the authors inserted all suggestions performed by the reviewer, that contribute to improving the quality of the manuscript.